# A Transect Through the Living Environments of Slovakia’s Roma Population: Urban, Sub-Urban, and Rural Settlements, and Exposure to Environmental and Water-Related Health Risks

**DOI:** 10.3390/ijerph22070988

**Published:** 2025-06-23

**Authors:** Lukáš Ihnacik, Ingrid Papajová, Júlia Šmigová, Mark Brussel, Musa Manga, Ján Papaj, Ingrid Schusterová, Carmen Anthonj

**Affiliations:** 1Institute of Parasitology, Slovak Academy of Sciences, 040 01 Košice, Slovakia; ihnacik@saske.sk (L.I.); papaj@saske.sk (I.P.); bystrianska@saske.sk (J.Š.); 2Department of Epidemiology, Parasitology and Public Health Protection, University of Veterinary Medicine and Pharmacy in Košice, 040 01 Košice, Slovakia; 3Faculty of Geo-Information Science and Earth Observation, International Institute for Geo-Information Science and Earth Observation (ITC), University of Twente, 7500 AE Enschede, The Netherlands; m.j.g.brussel@utwente.nl; 4The Water Institute at UNC, Department of Environmental Sciences and Engineering, Gillings School of Global Public Health, University of North Carolina at Chapel Hill, Chapel Hill, NC 27599, USA; 5Department of Electronics and Multimedia Telecommunications, Technical University of Košice, 040 01 Košice, Slovakia; jan.papaj@tuke.sk; 6Faculty of Medicine, East Slovak Institute of Cardiovascular Diseases, 040 01 Košice, Slovakia; ischusterova@vusch.sk

**Keywords:** endoparasites, health inequalities, one health, Roma communities, social inequalities, vulnerable groups, WASH

## Abstract

The Roma population is one of Europe’s largest ethnic minorities, often living in inadequate living conditions, worse than those of the majority population. They frequently lack access to essential services, even in high-income countries. This lack of basic services—particularly in combination with proximity to (stray) animals and human and solid waste—significantly increases environmental health risks, and leads to a higher rate of endoparasitic infections. Our study sheds light on the living conditions and health situation in Roma communities in Slovakia, focusing on the prevalence of intestinal endoparasitic infections across various settlement localisations. It highlights disparities and challenges in access to safe drinking water, sanitation, and hygiene (WASH) and other potentially disease-exposing factors among these marginalised populations. This study combines a comprehensive review of living conditions as per national data provided through the Atlas of Roma communities with an analysis of empirical data on parasitological infection rates in humans, animals, and the environment in settlements, applying descriptive statistical methods. It is the first study in Europe to provide detailed insights into how living conditions vary and cause health risks across Roma settlements, ranging from those integrated within villages (inside, urban), to those isolated on the outskirts (edge, sub-urban) or outside villages (natural/rural). Our study shows clear disparities in access to services, and in health outcomes, based on where people live. Our findings underscore the fact that (i) place—geographical centrality in particular—in an already challenged population group plays a major role in health inequalities and disease exposure, as well as (ii) the urgent need for more current and comprehensive data. Our study highlights persistent disparities in living conditions within high-income countries and stresses the need for greater attention and more sensitive targeted health-promoting approaches with marginalised communities in Europe that take into consideration any and all of the humans, ecology, and animals affected (=One Health).

## 1. Introduction

Globally, nearly 4.4 billion people still lack access to safe drinking water, and more than 1.7 billion people lack basic sanitation services, a fourth of whom continue to defecate in the open [1], despite the major efforts of global institutions such as the World Health Organisation (WHO) and UNICEF, and the United Nations Agenda’s Sustainable Development Goal 6 (SDG 6), to ensure *availability and sustainable management of water and sanitation for all* [2,3]. According to 2023 data from the Joint Monitoring Programme (JMP) [4], which regularly produces estimates of global WASH progress, most countries and people without safe drinking water, sanitation, and hygiene (WASH) services are located in the so-called Global South, primarily in low- to middle-income countries. However, large pockets of people facing such challenges also exist in high-income countries, e.g., in Europe. One such group are Roma people, an ethnic group originating from northern India who have traditionally led a nomadic lifestyle and have settled in various countries in Europe. Roma people form Europe’s largest ethnic minority (numbering about 1.35% of the total population) [5], mainly residing in the southeastern part of Europe. The term “Roma” encompasses diverse and heterogenous groups, such as Gypsies, Travellers, Manouches, Ashkali, Sinti, and Boyash [6]. According to European Parliament statistics, there are approximately 10 to 12 million Roma people in Europe [7,8].

Roma communities often face significant disadvantages compared to the majority populations, including discrimination, poverty, and social exclusion. Inequalities in access to health care, inadequate living conditions, and stigmatisation contribute to their exclusion [9,10]. Additionally, the lack of comprehensive data on Roma individuals, communities, and living conditions hinders an accurate understanding of their socio-economic and living situation [8,11]. The poor sanitation, crowded living conditions, lack of access to drinking water, and low socioeconomic status—particularly in combination with proximity to (stray) animals and human and solid waste—prevalent in many Roma settlements significantly impact their health and well-being while exposing them to endoparasitic diseases [11]. Endoparasites thrive under these exact conditions, and this highlights the urgent need for improved living standards [12].

Globally, intestinal parasitic infections caused by protozoans like *Giardia duodenalis* and helminths such as *Ascaris lumbricoides* and *Trichuris trichiura* are widespread in places where precarious living conditions and substandard WASH coincide. It is estimated that 3.5 billion people worldwide are affected by intestinal parasitic infections, with 450 million experiencing symptomatic illnesses and over 200,000 annual deaths [13]. Infections with these parasites often lead to various health problems. Early symptoms like diarrhoea or vomiting can progress into severe gastrointestinal infections, leading to poor nutrient absorption, dehydration, persistent watery diarrhoea, anaemia, and even malnutrition. In children, the risks are even greater, as these diseases can impair growth (stunting), cause intellectual delays, and negatively affect cognitive and learning abilities [14]. However, despite the severity of endoparasitic infections, their prevalence among the Roma communities is unknown due to a lack of comprehensive and reliable statistical data on their health status [15].

In Slovakia, according to the Atlas of Roma Communities 2019 (from now on also referred to as “the Atlas”), there are approximately 440,000 Roma people [15]. Despite Slovakia being a member of the European Union with a relatively high standard of living, the country still faces significant challenges related to social exclusion and inadequate access to essential services for its marginalised groups. Most of the Roma people live in concentrated settlements, often pushed to the outskirts of villages and isolated from the majority population by natural barriers like forests and rivers or by man-made structures such as fences, train tracks, waste collection sites, and factories. Yet, the structure—and centrality—of Roma settlements is very heterogeneous, ranging from localisation within villages (inside, urban), to those isolated on the outskirts (edge, sub-urban) or outside villages (natural/rural) [16].

Living conditions in these settlements are often inadequate, with low personal and environmental hygiene standards. Other contributing factors to this situation are the lack of education in Roma communities and the aforementioned social exclusion [12,15]. Overall, nearly one-third of the inhabitants of these communities live in illegal dwellings and on unsettled lands, in wooden huts, shacks, or other substandard dwellings, which offer significantly lower housing quality compared to legally constructed homes. In nearly all of the dwellings in Roma settlements, overcrowding is a common issue, with an average of eight people per dwelling [15], more than three times higher than the average density per dwelling in Slovakia (2.5 people per dwelling) [17]. This overcrowding negatively impacts the quality of housing and the health of the population. Overcrowding is not the only challenge in Roma settlements; poor infrastructure is also common. According to the summary in the Atlas [15], nearly 64% of inhabitants in Roma communities use the public water supply, while only 40% have access to sewerage systems. Although conditions in the settlements have improved since earlier editions of the Atlas, the living standards of the Roma remain insufficient.

Living conditions in Roma communities in Slovakia require locally informed, tailor-made interventions and strategies, and targeted decision-making regarding social inclusion and the use of WASH utilities. However, this is often challenging due to the lack of current and comprehensive information about these groups in Slovakia. The Atlas partially addresses these needs in Slovakia [15], but its usefulness is limited by its availability only in Slovak, making it less accessible to scientists from other countries working with this topic. While the Atlas offers extensive information and some comparisons of basic living conditions, it lacks a comprehensive comparison of living conditions focusing on water and sanitation across different types and localisation of settlements.

With our study, we aim to

provide deep insights into the living environments of Slovakia’s Roma population;compare living conditions in settlements at different levels of centrality, including urban, peri-urban, and rural;analyse the occurrence of endoparasite diseases in different Roma settlements across Slovakia, thereby offering a comprehensive view of the overall health of people, animals, and the living environment.

This is the first comprehensive insight into and assessment of the diverse living environments of one of Europe’s ethic minority groups, and the first integration of desk study knowledge and parasitological data in Slovakia and beyond. It sets the stage for future research and evidence-informed public health interventions that consider the needs of Roma communities while considering their complex human–animal and environment interactions.

## 2. Methods

This study relies on theoretical and conceptual framing in the context of medical and health geography and public health. A short introduction to medical and health geography and to the relevance of place—or localisation, as we frame it here—for health, as well as to One Health, is therefore provided in the following.

### 2.1. Conceptual Framing

The linkages between environment and public health underlie spatiotemporal dynamics, and vary in space and time. Geography, according to Waldo Tobler (1970) [18], relates everything to everything else [18]. To understand health challenges at the interface of environment and public health, the use and application of medical and health geography, sub-disciplines of geography, is helpful. The investigation of relationships between health and diseases—in this present study specifically environmental health risks and endoparasitic infections—in space and time relies on links between environmental factors (e.g., topographical, hydrological, environmental hygiene, presence of animals), population and governance factors (population, infrastructure, cultural context, policy, economy), water (in)security parameters (e.g., sanitation and sewerage management—or lack thereof), and factors of exposure to disease [19]. Medical and health geography approaches can bridge different health- and health system-related contexts and scales.

Health inequalities (in the exposure to or prevention of disease) can be very place-specific as they are determined to a significant extent by the resources—some with positive, some with negative valence—to which individuals have access. These place- or neighbourhood-based resources are not equally relevant for all population groups, but can play a key role for the more place-bound population groups [20,21]: those, in particular, who are more dependent on local resources such as individuals with reduced access to private transportation, individuals in poor health, or people with low incomes [22]. This also applies to individuals who spend more time in their locality. Centrality plays a major role in accessing such resources; thus, centrality also matters in terms of disease exposure and health promotion.

Another concept of high relevance to our study is One Health, previously defined as “a narrow approach combining public health and veterinary medicine” or “a wide approach including both scientific fields, core concepts, and interdisciplinary research areas.” In both cases, the focus is usually on safeguarding the health of vertebrates, while ecosystems are also included in the model [23,24]. One health is closely linked to WASH [25].

### 2.2. Methodological Approach

This study combines two methodological approaches. The first is a desk study analysing the living conditions of Roma communities in various settlement localisations using information from the ARC [15]. The second examines the rate of parasitological infection in the human population, animals, and the level of environmental contamination in selected Slovak settlements, utilising empirical results from our own previous parasitological studies—some of which were published [11], and others of which were not.

### 2.3. Desk Study

#### 2.3.1. Data Collection

Information on Roma people and settlements was obtained from the Atlas [15]. It includes data from 825 municipalities, which represent nearly 29% of the overall number of municipalities in Slovakia, with a total of 1102 different Roma settlements. As its focus is primarily on municipalities with at least 30 Roma inhabitants or where Roma people constitute 30% of the population, not all municipalities are covered in the Atlas. The Atlas contains data on living conditions, (WASH) infrastructure, service availability, and spatial distance of Roma settlements from other populations. These data were collected for the period from July 2018 to September 2019.

Factors included in our overview and analysis were selected from the Atlas [15] to cover aspects related to health promotion or health risk potential [9,11,26]. We also took into account whether data were available for each settlement in the Atlas, as some factors had missing information. In total, we selected 21 different factors (number of people; density of people; usage of public water supply; personal wells; public wells or taps; non-standard water source; usage of public sewage; personal cesspool/sumps (liquid waste reservoir); home wastewater treatment plants; information about ownership of waste receptacles; information about floods; other types of natural disasters; distance to general practitioners; distance to paediatricians; distance to bus stop; distance to train stop; usage of electricity; gas distribution system; home ownership; house types; density in dwellings). We hypothesised that access to WASH services, utilities, public transit stops, and proximity to educational and health facilities, along with associated health risks, would vary depending on the centrality of a settlement. To analyse these differences, we categorised the data into three distinct settlement localisations according to Ravasz et al. [12]:*Inside* settlements, located within the main village core, including streets, apartment buildings, and yards that host more than 30 Roma inhabitants but remain an integral part of the village’s development area.*Edge* settlements, situated on the outskirts of the village core, where houses in the main part of the village gradually transition to the periphery, forming a Roma concentration immediately adjacent to the village’s development area.*Outside* settlements, located outside the village core, defined as concentrations separated from the main village development area by undeveloped land, railway lines, rivers/streams, or roads.

The localisation of Roma settlements related to the main village development was defined for each settlement in the Atlas and is based solely on the geographical position of the Roma settlement within the village boundary. These three different types of settlement localisation also represent distinct levels of centrality. *Inside* settlements often have properties similar to those of urban environments due to their central characteristics. *Edge* settlements show features of both villages and cities, resembling sub-urban environments. *Outside* settlements are often near nature and resemble a village or an isolated concentration. They have properties similar to rural or natural environments.

Selected factors were subsequently grouped into eight bigger categories to describe features of the living conditions, infrastructure, and service availability. These categories included information about population, water sources, wastewater and waste disposal, flooding, distance to medical facilities, infrastructure, housing, and energy (Table 1).

#### 2.3.2. Data Analysis

First, we calculated the average values for individual factors for all settlements used in the Atlas to establish a basis for comparison of different Roma living conditions in different localisations of settlements. Additionally, descriptive statistics were calculated for each factor in one of the three types of settlement localisation mentioned above. By analysing measures of central tendency, skewness, and dispersion of data, we obtained an overview of the conditions in the settlements. Next, we established a logical scale to categorise the percentage of dwellings using utilities. The values were divided into five equal intervals: 0–20%, indicating very low; 20–40%, low; 40–60%, moderate; 60–80%, high; and 80–100%, indicating a very high number of dwellings connected to the utility. Frequencies of settlements for each factor were calculated, providing insights into how many settlements have a certain degree of dwellings connected to utilities, which was compared with other settlement localisations. We categorized the factors related to distance of settlements from medical and educational facilities and public transport stops into five different-sized classes: 0–1 km (within the settlement), 1–3 km (likely within the village), 3–7 km (possibly in the village or a neighbouring village), 7–15 km (likely in the nearest town or large village), and more than 15 km (located in a larger regional city). Frequencies of settlements for each factor were also calculated. This approach facilitated a clear understanding of the percentage of settlements with facilities or stops at specific distances. The distance classes were chosen based on the overall average, as well as minimum and maximum distance values for each factor. Additionally, data skewness was considered, with most factors showing a positive skewness. Consequently, the division into five classes, where four classes were within 0–15 km, plus one class for distances greater than 15 km, was selected to accurately reflect the distances. Lastly, factors regarding the occurrence of natural disasters were classified into two categories: “Yes”, if the settlement had experienced the occurrence of some kind of natural disaster, and “No”, if it had not. Frequencies of settlements for each factor were calculated. As the number of settlements in the three localisations differed, frequencies were converted to percentage values for graphical representation and comparison. The percentage of frequencies was calculated by dividing the number of settlements with a certain level of utility usage by the total number of settlements in one settlement’s localisation.

### 2.4. Parasitological Analysis

#### 2.4.1. Data Collection

In assessing the health of inhabitants and environmental contamination, we gathered epidemiological data on intestinal helminthiasis occurrence. All the samples used in this research were collected over the five years from 2019 to January 2024, with some of the results already published [12]. Published datasets were selected based on their geographic relevance and demographic comparability to the current study population. All datasets were standardised in format prior to analysis, enabling integrative comparison and interpretation of parasitological data This included parasitological data from humans, animals (particularly dogs), and the environment. Incorporating these diverse data allowed for a comprehensive assessment of human and environmental health across living environments (rural, edge, and central). In total, we gathered information on 5378 human stool samples from 161 locations in eastern Slovakia, collected randomly and voluntarily, or with the help of local doctors, paediatricians, and social workers. Additionally, we gathered information on 2764 dog faecal samples from 41 locations and 710 soil samples from 31 locations. For each human stool sample, informed consent was obtained from all participants before data collection. Participants received information containing the study’s purpose, the nature of participation, and the anonymity of data. Participants were assured that their involvement would not affect their privacy or their rights. The informed consent, together with the study protocol, was reviewed and approved by the Ethical Commission of Košice Self-governing Region, following international ethical guidelines for research involving human subjects (document no. 5436/2019/ODDZ-25820). Collected data were stored on physical drives and accessed only by authorised persons. Before the analysis, the data were anonymised and generalised so that any sensitive information was not accessible.

Samples were sorted by location based on information from informed consent or parasitological collection notes. Human stool samples were categorised differently, firstly into samples from non-Roma people and Roma people, which were later categorized by settlement localisation (*inside*, *edge,* and *outside*). Similarly, dog and soil samples were grouped into *inside*, *edge*, and *outside* settlements. This categorisation was based on the sample locations from our research. The final determination of settlement localisation was made with the help of the Atlas [12], which provides detailed information on settlement localisation in villages. This allowed us to accurately assign samples to their respective settlement localisation by cross-referencing our notes with the Atlas [12].

#### 2.4.2. Data Analysis

All gathered samples were analysed within 24–48 h at the Institute of Parasitology of the Slovak Academy of Sciences in Košice. Human stool samples were analysed either by a commercially available kit (Paraprep L; DiaMondiaL, Vienna, Austria) or with the SAF method. Dog faecal samples were analysed using the classic flotation method with Shaeter’s flotation and the Faust flotation solution. Soil samples were analysed by the sedimentation–flotation method as described by Kazacos [27]. After parasitological analysis, epidemiological statistics were calculated. Odds ratio (calculated as odds of finding endoparasitic eggs in samples from one group divided by the odds of endoparasitic eggs being present in the other group), percentage positivity, chi-square, *p*-value, and confidence intervals were calculated from all samples and for all of the three settlement locations using MS Excel (Microsoft Office LTSC Professional Plus 2024; version 2408, Redmond, WA, USA) and RStudio software (Posit PBC; version 025.05.1+513, Boston, MA, USA).

## 3. Results

### 3.1. Transect Through Roma Living Conditions

Drinking water access was found to be highest in *inside* settlements, with over 70% of dwellings connected to the public supply, 17% using personal wells, and 7% relying on public wells. In *edge* settlements, 60% of dwellings use the public water supply, 20% use personal wells, 13% rely on public wells, and 7% use non-standard sources. The lowest level of access exists in *outside* settlements, resulting in 8% using non-standard sources like rivers or streams (Table 2).

*Inside* settlements have 50% of dwellings using public sewerage and 23% using cesspools. In *edge* settlements, 26% use public sewerage and 25% cesspools. *Outside* settlements have 15.5% connected to public sewerage and 28.5% using cesspools (Table 2).

*Inside* settlements face fewer natural disasters (16% facing floods). *Edge* settlements are most vulnerable (33% experiencing floods), while *outside* settlements are at moderate risk (25% facing floods) (Table 2).

Health facilities are located closest to *inside* settlements, with GP offices about 1.8 km and paediatricians 3.6 km away. In *edge* settlements, GPs are 4 km and paediatricians 5.6 km away. GPs and paediatricians are farthest from *outside* settlements (5 km and 6.5 km away respectively) (Table 2).

Bus stops are generally within 1 km of settlements, but train stops are farther, averaging 6 km in *inside* settlements and 8–9 km in *edge* and *outside* settlements (Table 2).

*Inside* settlements have up to 2 unapproved house structures, while *edge* settlements have more unapproved structures on average (7 to 8 brick houses, up to 1 wooden house, and 5 to 6 unapproved huts). *Outside* settlements have the highest number of unapproved buildings (7 brick, 2 to 3 wooden houses, and 10 to 11 huts) (Table 2).

Utility access also varies, with electricity decreasing from 93% in *inside* settlements to 80% in *outside* settlements. Municipal gas use drops from 30% in *inside* settlements to 4% in outside ones (Table 2).

### 3.2. Living Conditions of Roma People in Settlements at Different Levels of Centrality

To better understand the living conditions across settlement localisations at different levels of centrality, we compared the percentage of settlements with varying connectivity of dwellings to utilities or the percentage of settlements with different distances from facilities.

As shown in Figure 1, up to 46% of *outside* settlements have less than 20% of dwellings connected to the public water supply, while almost 40% of these settlements have nearly all dwellings connected. In *inside* settlements, about 18% of settlements have up to 20% of the dwellings connected. *Edge* settlements fall somewhere in between, showing decreasing connectivity as distance from the municipality increases. Personal and public wells are most common in *outside* settlements, used by 19% of these settlements, and 13% have almost all dwellings relying on public wells. In comparison, only 10% of *inside* settlements have nearly all dwellings using personal or public wells (3%), respectively. Across all localisations, 70–80% of settlements have few or no dwellings using personal wells. Similarly, 73% of *outside* settlements and 88% of *inside* settlements have dwellings with almost no connections to public wells. Over 90% of *inside* settlements have dwellings with minimal or no use of non-standard water sources. Similarly, more than 90% of *edge* settlements have few dwellings using non-standard water sources, but in 4%, nearly all dwellings rely on non-standard water sources. In *outside* settlements, only 2% have nearly all dwellings using non-standard sources (Figure 1).

Public sewage is used in almost all dwellings in 42% of the *inside* settlements. Only 9.8% of *outside* settlements have most dwellings connected to sewage, while nearly 80% have none or very few dwellings connected to sewage, showing a decreasing gradient. In contrast, cesspool use increases, with 13% of *inside* settlements having all dwellings using cesspools, compared to nearly 20% of *outside* settlements. Almost all settlements in all localisations have very few or no dwellings connected to private wastewater treatment plants (Figure 1).

*Edge* settlements experienced natural disasters most commonly, with floods in 33% and other natural disasters in 23% of settlements. *Inside* settlements were less affected, with floods in about 16% and other natural disasters in only 9% of settlements. *Outside* settlements faced floods in 25% and other natural disasters in 16% of settlements (Appendix A).

Only 70% of *inside* villages, 40% of *edge* villages, and 22% of *outside* villages have a general practitioner’s office within 1 km. Only 13% of *inside*, 27% of *edge*, and 35% of *outside* settlements have such a facility within 3 to 7 km, while 3% of *outside* settlements must travel more than 15 km for adult healthcare. Paediatricians’ offices are even further away. Only 57.8% of *inside*, 32% of *edge*, and 17% of *outside* settlements have a paediatrician within 1 km. More settlements have paediatricians within 3 to 7 km, with 5% of *inside*, 6% of *edge*, and 8% of *outside* settlements having a paediatrician’s office more than 15 km away (Appendix A).

Almost all settlements have a bus stop within proximity, typically within 1 km. For *inside* settlements, approximately 98% have a stop within 1 km, and around 2% within 3 km, with very few settlements farther away. *Edge* and *outside* settlements show similar patterns, with 92% and 86.6% of settlements having a stop within 1 km, nearly 4% and 10% within 3 km, and 4% and 3% within 7 km, respectively. However, very few settlements are more than 7 km from a bus stop. Train stops are generally more distant. About 49% of *inside* settlements have a train stop within 1 km, compared to 30–35% for *edge* and *outside* settlements. Over 10% of inside, 18% of edge and 17% of *outside* settlements have a train stop more than 15 km away (Appendix A).

Unapproved houses are divided into three main groups: brick, wooden unauthorised built houses or additions, and huts (unauthorised shacks, built with a variety of materials). We divided the number of houses into intervals: from 0 to 5 (low), 5–10, 10–15, 15–20, 20–30, and over 30 (very high). These houses lack official permits and are often built on land not owned by the residents, so they are not recognised by the municipality. Unauthorised wooden houses are the least common, with up to 5 found in 99% of *inside* settlements and 95% of *edge* and *outside* settlements. Unauthorised brick houses are more prevalent, with over 30 found in 1% of *inside* settlements, 7% of *edge* settlements, and 6% of *outside* settlements. Over 30 unauthorised huts are present in nearly 7% of *outside* settlements, compared to only 0.2% of *inside* settlements. Overall, *outside* settlements have the most unauthorised dwellings, with 10% having more than 20 unauthorised brick houses, 3% having more than 20 unauthorised wooden houses, and 12% having more than 20 unauthorised huts (Appendix A).

We divided the dwellings into five groups based on the percentage of dwellings that use utilities. As shown in Figure 1, electricity grid connectivity is high, with nearly all dwellings connected to the grid in 85% of *inside* settlements, 74% of *edge* settlements, and 69% of *outside* settlements. On the contrary, almost no dwellings are connected in 3% of the *inside* settlements, nearly 8% of *edge* settlements, and up to 11% of *outside* settlements, showing a decreasing gradient of connectivity with increasing settlement isolation (Appendix A).

Gas network connectivity is generally poor, with settlements and many municipalities in Slovakia completely lacking this infrastructure. Most of the dwellings are not connected to the gas network in 63% of *inside* settlements, 79% of *edge* settlements, and 94% of *outside* settlements. Consequently, most dwellings use solid fuels like wood for cooking and heating, as just 19% of *inside* settlements, 7% of *edge* settlements, and 1% of *outside* settlements have nearly all the dwellings connected (Appendix A).

### 3.3. Parasitological Results

Overall, 5378 human stool samples were examined for the presence of endoparasite eggs. Parasitological analysis revealed that intestinal parasites were present in 8.9% of samples. We found that the highest positivity rate was in the environment of Roma settlements. *Outside* settlements had the highest positivity rate, with over 25% of samples testing positive. *Inside* settlements had a positivity rate of over 20%, while *edge* settlements had the lowest rate at 14.3% (Figure 2). The most prevalent were the eggs of helminths, *Ascaris lumbricoides* and *Trichuris trichiura*, as well as the oocyst of protozoan *Giardia duodenalis*. Although we did not perform perineal swap analysis, we also detected a few *Enterobius vermicularis* eggs in stools. The odds of parasitic infection varied by settlement localisation. The highest odds ratio (OR) for infection was found in *outside* settlements (OR 1.72), compared to other settlements and the majority population. *Inside* settlements had a lower OR of 1.17, while *edge* settlements had the lowest OR at 0.56. For comparison, the majority population had the fewest positive samples, with an OR of 0.02 (Appendix A).

A total of 2764 samples of dog faecal samples and 710 soil samples were analysed for the presence of eggs of intestinal endoparasites. The analysis revealed that 41.7% of dog faeces samples and 35.2% of soil samples were positive for the presence of parasite eggs. The highest prevalence of infection in dog samples was observed in dogs from *outside* settlements, with a rate of 61.8%. Dogs from *edge* settlements had a lower prevalence of 49.1%, while dogs from *inside* settlements had the lowest prevalence at 34.6%. In dog samples, the eggs of the family Ancylostomatidae, *Ascaris* spp. and *Toxocara canis* were the most prevalent. The presence of *Ascaris* spp. was surprising, as they are not typical dog parasites. Soil samples showed the highest prevalence of parasite eggs in *edge* settlements at 70.5%, slightly lower in *outside* settlements at 57.4%, and significantly lower in *inside* settlements (11.6%) (Figure 2). Eggs of *Toxocara* spp. and *Trichuris* spp. were the most prevalent. The probability of infection in dogs was highest in *outside* settlements (OR 2.61), decreased in dogs from *edge* settlements (OR 1.4), and was lowest in dogs from *outside* settlements (OR 0.44). For soil samples, the highest probability of contamination was found in *edge* settlements (OR 8.33), lower in *outside* settlements (OR 3.05), and lowest in *inside* settlements (Appendix A).

## 4. Discussion

### 4.1. Filling Knowledge Gaps on Water and Health Insecurity in Different Roma Living Environments

To date, only a few studies from Europe, each with a different focus, have emphasised the living conditions, limited access to essential services, and the health status of Roma communities. This is the first study to comprehensively examine the diversity and complexity of Roma people’s living environments, based on the case of Slovakia, shedding light on the unique health risks associated with these varied conditions, while also incorporating information on endoparasitic diseases across different levels of centrality. Settlements located outside urban areas have particularly precarious conditions, with less than half of the dwellings having access to safe wastewater disposal, resulting in unsanitary living conditions and environmental degradation. Additionally, settlements located outside village centres are more vulnerable to natural disasters, which often devastate the already limited infrastructure in these areas, posing a dual risk to health. Large household sizes in small homes create conditions that further destabilise already weak living standards [28]. In addition, Roma communities face several challenges closely tied to their cultural practices and significantly different demographic behaviour compared to the majority population. Many challenges faced by Roma communities stem from their segregation into settlements with often inadequate living conditions. Rural Roma settlements (*outside* settlements), in particular, suffer from a worse social reputation due to their isolation, lack or inefficiency of infrastructure, and higher levels of poverty compared to urban settlements. These rural areas typically experience limited access to essential infrastructure services such as clean water, sanitation, healthcare, and education, which perpetuates negative stereotypes and deepens social exclusion. Often, these settlements are overlooked in decision-making processes by authorities, receiving attention only in critical situations or when laws need to be enforced. Existing health services in Slovakia are often physically distant, under-resourced, and not culturally adapted to meet the needs of Roma populations. Although Slovakia has an epidemiological information system that monitors the occurrence of various diseases, it primarily focuses on individuals who have developed clinical symptoms and sought medical attention [29]. In the case of intestinal diseases, individuals may be infected but exhibit mild or no symptoms, and the disease might go unnoticed and unreported and potentially spread further. Thus, surveillance systems do not adequately capture parasite prevalence in these settings, leaving many infections undiagnosed and untreated—and the communities at risk of infection. This highlights a critical gap in public health efforts, emphasising the need for authorities to shift from partial responses to more comprehensive, sustained, and inclusive strategies for parasite control and environmental health protection. Ultimately, this marginalisation can perpetuate negative stereotypes and deepen social exclusion. Addressing these issues requires targeted interventions, improved access to healthcare, and concerted efforts to reduce social and environmental disparities [26].

### 4.2. Root Causes of Water and Health Insecurity Among These Marginalised Groups

Settlements are often geographically isolated from the majority population and located near industrial zones, waste disposal sites, or agricultural cooperatives, disconnected from main roads and transportation services [30]. This isolation limits access to employment opportunities, education, professional training, and essential health services, while further exacerbating the challenging living conditions [11,31]. As our results suggest, living conditions within Roma communities can differ substantially depending on geographic location, and socio-economic and cultural context. Access to essential services such as drinking water, electricity, basic sanitation, and transportation is fundamental for ensuring well-being and quality of life. Improving living conditions in Roma settlements is hard to achieve without access to essential services, employment, and opportunities for skill development [26]. This is further compounded by their limited participation in decision-making and the lack of political will to address their needs. As a result, efforts to enhance Roma communities remain ineffective, perpetuating cycles of poverty and exclusion [11]. The scarcity of formal employment opportunities further continues this cycle, as few jobs provide essential benefits like health insurance, maternity leave, and pensions, deepening their economic and social marginalisation [7]. Living condition deficiencies are exacerbated by existing natural and social barriers. Home ownership is a very critical factor too, impacting access to utilities, since houses without proper authorisation are not officially recognised by authorities and therefore not connected to utilities. In addition, proximity to public transport stops is crucial for the accessibility of essential services like healthcare and schools, especially when these services are located away from the settlement. With increasing distance from the centre to the village, proximity to public transport decreases, and the willingness of inhabitants to travel reduces. In contrast, shorter distances within settlements closer to the village centre reduce travel times, improving access to necessary services. The location of settlements plays a significant role in the aforementioned challenges among marginalised groups. Settlements located outside or on the edge of the villages often face significant barriers to accessing essential services and infrastructure [15]. The isolation of such settlements exacerbates these issues, leading to increased vulnerability to parasitic diseases, poor health outcomes, and broader social exclusion. Addressing these location-based disparities is essential for improving overall water and health security in marginalised communities.

There is also a pressing need for targeted training programs to improve the skills, employability, and inclusion of Roma individuals. Such initiatives could help bridge the gap between these marginalised communities and the broader labour market, providing them with the tools they need to secure stable and meaningful employment and better living conditions.

### 4.3. WASH and Resulting Health Insecurity in Roma Settlements

Settlements have low access to safe WASH, with decreasing access as the level of centrality reduces, and this is the most important factor in the occurrence and spread of endoparasites [9]. This finding is in agreement with previous evidence on the effect of geographic location on WASH conditions. More central locations, going from rural to semi-urban to urban, tend to have better conditions [32,33,34]. Generally, this is based on a variety of factors, including the availability and quality of different infrastructure, particularly housing conditions, water supply, sanitation and public waste management, and how these interact, supporting previous research. They also depend on many other factors such as how communities function, and whether they are marginalised by government and other stakeholders [34]. In addition, environmental conditions such as the occurrence of flooding and other natural disasters challenge water and health insecurity further, as does limited access to healthcare when needed [11]. Our analysis shows that *inside* settlements experience fewer disasters compared to *edge* settlements, where nearly one-third of settlements face floods or other disasters. *Outside* settlements face moderate risks. These disasters worsen conditions by damaging already inadequate sanitation systems and increasing environmental contamination, further contributing to the spread of endoparasites and lowering living standards overall.

According to our analyses, Roma inhabitants in settlements *outside* villages have the highest disease burden, where over 25% of samples were positive for the endoparasite propagative stages. In contrast, settlements *inside* villages had endoparasitic eggs in over 20%, and *edge* settlements had the lowest rate at 14%. The higher prevalence of endoparasitic infections is primarily attributed to the poor living conditions of Roma populations residing in substandard settlements on the outskirts or outside of villages [35]. The higher number of intestinal parasitic infections found in settlements in our study points to the fact that contamination through the faecal–oral route is a well-known transmission pathway not only for intestinal parasites [36]. Less than half of dwellings in *outside* settlements are connected to the public water supply, with a significant portion relying on wells or non-standard sources, and just 44% of the dwellings have safe wastewater disposal facilities. These unsafe water sources, inadequate sanitation, and poor hygiene practices often mediate this route in the isolated Roma settlements [15]. This contamination not only affects individuals directly but also contributes to environmental contamination and the spread of infection. Therefore, the findings underscore the critical need for public health interventions. Ensuring access to safe drinking water, effective waste disposal systems, and public health education is vital to interrupt the faecal–oral transmission cycle. Furthermore, these findings should be interpreted not only as evidence of the unevenness of sanitary conditions but also as reflections of the broader exclusion of the Roma community. The inadequate infrastructure observed in segregated Roma (*outside*) settlements is not accidental. It is the outcome of multiple factors, such as long-term institutional neglect, which is, however, unintentional, as municipalities often do not know how to approach these communities and their problems. This later leads to uneven management in municipalities, as attention is diverted from Roma communities, and their needs are often only addressed in emergency cases. This fact, along with the highest distance from health facilities, can result in contracting and delayed diagnosis and treatment of parasitic infections, contributing to higher prevalence rates in *outside* settlements (OR 1.72) compared to other settlements and the majority population. Higher positivity rates in samples from Roma individuals living in segregated settlements compared to other groups of people have been previously recorded [12,37]. For inhabitants, the primary sources of endoparasites are the contaminated environments where they live. These environments are polluted by the faeces of infected animals. To thoroughly address this issue, our research includes samples from both soil and dogs. Soil and dog samples are crucial as they can host part of the life cycle of parasites and can contribute to environmental contamination, thereby increasing the risk of infection for humans. Additionally, the risk of infection is heightened in areas where people frequently come into contact with wild animals, particularly in outside or edge settlements. This risk is further compounded by inadequate water and sanitation facilities, which limit effective hygiene practices and increase the likelihood of infection spread [11].

In soil, we found the most positive samples in *edge* and *outside* settlements, with 70% and 58% positivity, respectively. In contrast, only 12% of soil samples in *inside* settlements were positive. This difference can be attributed to several factors. Settlements which are further away from urban centres often have bad connectivity to or completely lack proper sanitation infrastructure, which leads to the disposal of untreated human waste directly into the environment, increasing soil contamination; or they are closer to the forests and fields, where the higher density of wild animals contributes to the frequent parasitic contamination of the soil. As Antolova et al. [38] confirmed, wild animals, especially foxes and stray dogs from Slovakia’s rural areas, have a higher prevalence of *Toxocara* spp., contributing to this elevated risk.

Both of these points are linked with the high prevalence of *Ascaris* spp. in dog faeces that we discovered. Typically associated with humans and wild animals, *Ascaris* spp. are rarely found in dogs, making this discovery a clear indicator of severe environmental contamination. Stray dogs, often forced into coprophagic behaviour due to food scarcity, are at increased risk of parasitic infections. The presence of *Ascaris* spp. in dog faeces suggests coprophagous behaviour in dogs and points to poor living conditions in these settlements. Eggs of *Ascaris* spp. primarily spread through faecal contamination in environments around human dwellings, often due to inadequate sanitation and open defecation [9]. This widespread environmental contamination, particularly evident in settlements outside villages, underscores the urgent need for better hygiene and waste management systems.

Our findings indicate that 50% and more than 62% of dogs from *outside* and *edge* settlements were infected, respectively. In contrast, just 37% of dogs from *inside* settlements were infected with endoparasitic propagative stages. This higher infection rate in dogs from more remote settlements is mainly due to their free roaming and interactions with contaminated environments and wild animals. Furthermore, the lack of veterinary care and control in these areas facilitates the spread of parasitic infections among the dog population [12]. Dogs in cities are in close contact with people and have become part of households, but they can become infected when they are in contaminated environments and this may contribute to the spread of infection further [37].

The continued prevalence of parasites in Roma people and their living environments reflects a broader failure of cash investments and long-term public intervention. Although over time some infrastructure improvements have occurred (when comparing newer editions of the Atlas from 2019 [15] with the Atlas of Roma Communities from 2013 [39]), these efforts have been uneven and poorly maintained. Public authorities often overlook segregated settlements, especially those without legal housing status, making it difficult to officially connect them to water or sewer systems. This exclusion perpetuates open defecation, poor waste management, and the use of non-standard water sources. All of these factors contribute to environmental contamination, which in turn directly contributes to faecal–oral parasite transmission. Unless these root causes are addressed, parasitic infections will remain, and public health interventions will remain reactionary rather than preventive.

### 4.4. Using Evidence for Targeted Interventions and to Improve Health-Promoting Planning

Substandard living conditions across different living environments challenge not only the health and well-being, but also the opportunities of Roma communities in Slovakia. Endoparasite infections are widespread due to the low levels of safe drinking water, sanitation, and hygiene. According to the official estimates by the JMP, 100% of urban and 99.6% of rural households in Slovakia have access to WASH [40]. Looking at the Atlas of Roma Communities [12], this is far from the reality (Table 2). To reduce health risks, the WASH situation in Roma settlements needs to be better understood, assessed, and monitored. Tools are needed that can capture WASH inequality at a sub-national level, and communicate it to decision-makers in the Ministries of Health, Ministry of Environment, and Ministry of Investments, Regional Development, and Informatisation of the Slovak Republic; the Regional Office of the World Health Organization in Europe; UNICEF; and the JMP, as a foundation for the development of targeted interventions and health-promoting planning in Slovakia and beyond.

To improve the overall living conditions in Roma communities, a multifaceted approach is needed that addresses housing issues, and the lack of basic services. It further needs to aim to reduce the marginalisation and isolation of communities—both in terms of accessibility of services, and in terms of social isolation and political participation. Using their local knowledge and experience to co-design local contextualised solutions could be a first step to better understand the lived realities and challenges faced by Roma communities; this has worked well with other marginalised population groups in Europe [41].

Social Innovation (SI) initiatives present one potential opportunity to improve the living conditions of Roma communities by strengthening social networks and fostering community cohesion. These initiatives focus on altering existing social structures to create new opportunities for collaboration and mutual support among community members [42,43]. By empowering individuals and encouraging collective action, SI initiatives can help break the cycle of marginalisation and pave the way for sustainable community development.

Regenerative housing programs offer a comprehensive solution to the challenges faced by Roma settlements. By focusing on the regeneration of housing, these programs address both social and environmental issues, contributing to progress on multiple dimensions of sustainable development, including the United Nations Sustainable Development Goals (UNSDGs) [44]. The implementation of such programs could lead to the construction of safe, legal, and sustainable housing, replacing unauthorised dwellings and improving the overall quality of life for Roma communities.

### 4.5. Limitations

Our study is limited by the cross-sectional nature of the data, which did not allow us to establish causality or seasonal variations. Furthermore, we focus on settlements, but want to acknowledge that while some Roma communities settle in localities that serve as permanent residences, others live in temporary accommodation such as wooden houses, self-made scrap metal shacks, tents, trailers, or other structures, and are less sedentary, all of which has implications for their access to WASH, and resulting health risks [30,45]. Moreover, it is essential to mention that the living environments of Roma communities are not simply limited to an *outside*, *edge*, or *inside* settlement categorisation. Rather, they represent a full spectrum of conditions that vary significantly based on location. This is a simplification used for ease of understanding, but in reality, the situation is far more complicated. It is also good to point out that the ARC excludes settlements with smaller Roma populations, leaving a segment of the community unrepresented in both the analyses and the focus of organisations. Although important drivers to water and health outcomes overall, we could not consider the role of weather, climate, time, social determinants, and governance in these communities due to lack of data. These unmeasured variables likely play a role in shaping the living conditions and health risks of these communities. A significant limitation of this study was the sampling process. One key challenge was the reluctance of some residents to participate, driven by distrust and fear toward our institution, despite repeated assurances that we held no enforcement power and that participation was voluntary and confidential. Finally, our study was limited by the one-time sampling, where each volunteer provided a sample only once. Multiple sampling strategies are very problematic, including both ethical concerns and the potential distrust towards scientific institutions, particularly in marginalised communities.

## 5. Conclusions

This study sheds light on the living conditions and health challenges faced by Roma communities in Slovakia, with a particular emphasis on the occurrence of endoparasitic diseases and lack of WASH in different localisations of settlements at different levels of centrality. The findings show clear disparities in access to services, and in health outcomes, based on where people live. Those in more isolated settlements tend to suffer the most, with poor access to basic necessities like running water, sanitation, and healthcare, which leads to higher rates of parasitic infections. These infections often result in intestinal problems and, especially for children, stunted growth. Our findings underscore the fact that (i) place—geographical centrality in particular—in an already challenged population group plays a major role in health inequalities and disease exposure, as well as (ii) the urgent need for more current and comprehensive data. This study highlights persistent disparities in living conditions within high-income countries and stresses the urgent need for greater attention and more sensitive targeted health-promoting approaches and action with marginalised communities that take into consideration any and all of the humans, ecology, and animals affected (=One Health). By including data on environmental conditions, access to basic infrastructure, and environmental as well as animal parasite contamination, this study provides a more comprehensive understanding of the health risks in marginalised communities. The findings reveal not only human health challenges but also their interaction with environmental exposures and zoonotic risks, underscoring the importance of inclusive and public health interventions.

Improving access to essential services and providing better education and skills training are crucial steps toward breaking the cycle of poverty and disease in these communities. Future research should focus on collecting more detailed and up-to-date data on parasitological health and living conditions in Roma communities, while exploring broader environmental factors such as soil type, climate, seasonality, animal data (stray dogs in particular), or the social determinants that influence health in these areas. As this study provides only a snapshot of the current situation, it cannot determine how changes in living conditions over time influence health both directly and indirectly. Therefore, in future more longitudinal studies are needed to establish causality between living conditions and health outcomes. Tracking communities over extended periods of time would offer a deeper understanding of how specific factors impact health, including the prevalence of parasitic infections. Moreover, future research needs to consider local knowledge related to water and health insecurity more.

Our study lays the groundwork for more effective interventions aimed at improving the well-being of Roma populations in Slovakia. Interventions must prioritise infrastructure development to provide clean water and sanitation for all. Educational programs on hygiene are key to reducing infections. Policies should be tailored to the specific needs of each settlement, ensuring no community is overlooked. Our results are not only relevant to Roma communities in Slovakia and beyond, but also to other marginalised groups, such as homeless populations, who often remain overlooked by the general public, and underserved by their governments [33].

## Figures and Tables

**Figure 1 ijerph-22-00988-f001:**
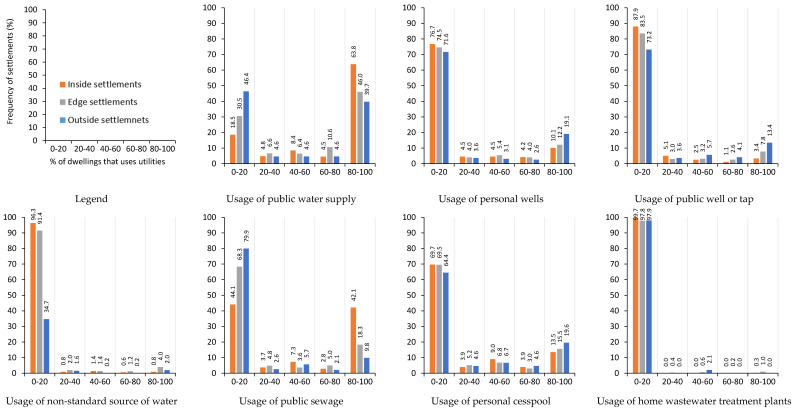
Water, sanitation, and hygiene situation in different Roma living environments at various levels of centrality calculated from Ravasz et al. [15].

**Figure 2 ijerph-22-00988-f002:**
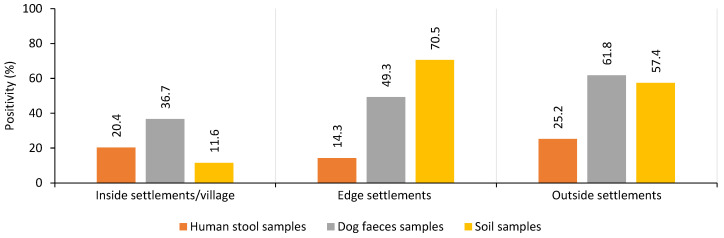
Comparison of the occurrence of endoparasites showed as percentage positivity between different Roma living environments and different types of samples in our sample localities.

**Table 1 ijerph-22-00988-t001:** Selected factors categorised with types of data obtained [15].

Category	Factors	Type of Data
Population	Number of inhabitants per settlement	Average number of people calculated from an interval of the estimate of inhabitants living in settlements
Water sources	Usage of public water supply	Percentage of dwellings that rely on each type of water source
	Usage of personal wells	
	Usage of public wells or taps	
	Usage of non-standard water source (nearby streams, rivers)	
Waste disposal	Usage of public sewage	Percentage of dwellings that rely on each type of waste and wastewater disposal
	Usage of a personal cesspool	
	Usage of home wastewater treatment plants	
	Ownership of waste receptacle	
Flooding	Floods	Percentage of settlements having previously experienced natural disaster
	Other natural disasters	
Distance to health care	Distance to general practitioners	Distance from settlement to facility in kilometers
	Distance to paediatricians	
Infrastructure	Distance to bus stop	Distance from settlement to facility in kilometers
	Distance to train stop	
Housing	Dwelling ownership	Number of houses with different ownership and house type
	House types	
	Density in dwellings	Average number of people per dwelling calculated from average number of people and number of dwellings
Energy	Household connectivity to the electricity grid	Percentage of dwellings in each settlements which relies on them
	Household connectivity to the gas distribution system	

**Table 2 ijerph-22-00988-t002:** Transect through different Roma living environments and living conditions at various levels of centrality calculated from Ravasz et al. [15].

Level of Centrality	Urban	Sub-Urban	Natural/Rural
Location of Settlements	Settlement Located Inside Village	Settlements Located at the Edge of Villages	Settlements Located Outside the Main Village
Map *	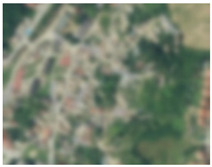	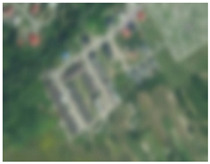	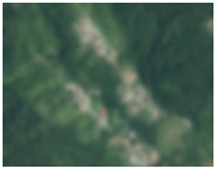
Images *	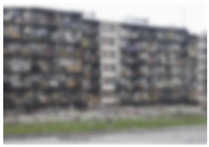	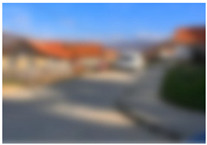	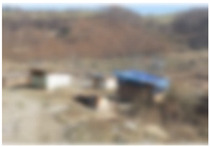
Water source	The majority of dwellings have access to safe and reliable water sources (>70% public supply).	60% of dwellings connected to public supply, around 20% use personal wells.	47% of dwellings are connected to public supply. 24% use personal wells, 21% public wells, 8% non-standard source.
Wastewater, waste disposal	Half of the dwellings are connected to public sewerage systems, 73% have safe wastewater removal. 84% have their own waste containers.	About 53% of dwellings have safe wastewater disposal. 72% of dwellings have waste containers.	44% of the dwellings have safe wastewater disposal. 57% of dwellings have their own waste containers.
Flooding	16% had experienced flooding events, and nearly 9% of settlements had faced other natural disasters.	33% had experienced flooding events, and 23% had faced other natural disasters.	25% of settlements had experienced flooding events, and 16% had faced other natural disasters.
Distance to healthcare	The average distance to the nearest GP is 1.8 km. The average distance to a paediatrician is 3.6 km.	The average distance to the nearest GP is 4 km. The average distance to a paediatrician is 5.6 km.	The average distance to a GP is 5 km. The average distance to a paediatrician is 6.5 km.
Infrastructure	The closest bus stop is only a few 100 m away. The closest train stop is about 6 km away.	The closest bus stop is only a few 100 m away. The closest train stop is about 9 km away.	The closest bust stop is within 1 km of the settlement. The closest train station is about 8 km away.
Housing	Average of 1–2 unapproved brick houses, 0–1 unapproved wood houses, 0–1 unapproved huts.	Average of 7–8 unapproved brick houses, 0–1 unapproved wood houses, 5–6 unapproved huts.	Average of 7 unapproved brick houses, 2–3 unapproved wood houses, 10–11 unapproved huts.
Energy	93% of dwellings are connected to electricity grid, 30% with access to gas distribution system.	85% of dwellings are connected to the electricity grid, 16% connected to the gas distribution system.	80% of dwellings connected to electricity grid, 4% connected to gas distribution system.

* The maps and images serve as examples illustrating the infrastructure and appearance of settlements at varying levels of centrality. Image blurring serves to ensure anonymity.

## Data Availability

Dataset regarding human stool samples, dog faeces samples and soil samples used in this publication are not publicly available because of safety concern for our participants and maintaining the anonymity of the area but they are available from the first author and the corresponding author upon reasonable request.

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
