# Peer review of "A Transect Through the Living Environments of Slovakia’s Roma Population: Urban, Sub-Urban, and Rural Settlements, and Exposure to Environmental and Water-Related Health Risks"

_ijerph, 2025, doi:10.3390/ijerph22070988_

Round 1
Reviewer 1 Report
Comments and Suggestions for Authors
This paper is based on a rich dataset offering a picture of WASH disparities across different Roma settlements in Slovakia. The policy relevance is clear and this is commendable. The inclusion of environmental and animal data adds meaningful depth through a One Health perspective.
That said, I think the manuscript would benefit from more engagement with critical public health literature. The concept of 'centrality' is especially interesting but feels underdeveloped. It could add more to the analysis with just a bit more reflection and explanation. Even though the paper is not aiming to be theoretical, some framing would help bring out the broader implications.
Referring repeatedly to “the Roma” risks flattening important differences within Roma communities in Slovakia, and the use of “Roma nationality” is misleading because Roma is an ethnicity. Some clarity and care in terminology would go a long way here.
On the methods side, it would be helpful to explain more clearly how the parasitological data, both published and unpublished, were selected and integrated. I also wondered how consent was handled, especially given the vulnerability of the communities involved. More detail on ethical practices would strengthen the paper.
The discussion section tends to repeat the findings rather than asking what they mean in a bigger sense. I would encourage the authors to reflect more critically (it does not need to be at length) on how health risks are shaped by exclusion, neglect, and uneven governance, especially in a country with the resources to do better.
Minor points: acronyms like ARC and JMP should be introduced earlier, and it would be good to acknowledge potential sampling bias.
This is an important paper and has valuable contribution.
Comments on the Quality of English LanguageThe English in the manuscript is generally good and understandable. It can benefit from some editing but I think the authors need to focus their attention on terminology, unclear phrasing, etc.
Author Response
Reviewer 1
Thank you for your suggestions on how to improve our manuscript, dear Reviewer #1. We answered all your comments and responded to every point as listed below. We particularly appreciate your suggestions to include a theoretical and conceptual framing, improve our discussion section and our conclusions.
Sincerely,
The Authors
R1: This paper is based on a rich dataset offering a picture of WASH disparities across different Roma settlements in Slovakia. The policy relevance is clear and this is commendable. The inclusion of environmental and animal data adds meaningful depth through a One Health perspective.
Authors’ response: Thank you for this really valuable feedback – while we are and were fully aware of the One Health concept applying to our research when drafting this article, we have not really highlighted that in our manuscript. We used your feedback as an incentive to highlight that much more, and incorporate the One Health concept more prominently in our abstract, conclusions and manuscript overall.
R1: That said, I think the manuscript would benefit from more engagement with critical public health literature. The concept of 'centrality' is especially interesting but feels underdeveloped. It could add more to the analysis with just a bit more reflection and explanation. Even though the paper is not aiming to be theoretical, some framing would help bring out the broader implications.
Authors’ response: We agree that incorporating and engaging more with critical public health literature strengthens our manuscript. Based on your comment, we added a new sub-chapter under our methods section where we refer to conceptual framing. Here, we briefly sketch medical and health geography, the concept of centrality, and One Health– all of which are vital to understand the broader implications of our work.
R1: Referring repeatedly to “the Roma” risks flattening important differences within Roma communities in Slovakia, and the use of “Roma nationality” is misleading because Roma is an ethnicity. Some clarity and care in terminology would go a long way here.
Authors’ response: Thank you for pointing out that the huge heterogeneity of within Roma communities remains hidden with our terminology used. We revised according to your suggestion, and included additional key references.
R1: On the methods side, it would be helpful to explain more clearly how the parasitological data, both published and unpublished, were selected and integrated. I also wondered how consent was handled, especially given the vulnerability of the communities involved. More detail on ethical practices would strengthen the paper.
Authors’ response: Thank you for this comment. We agree that more detail on the integration of published and unpublished data and ethical practices would strengthen the paper. Information on how the parasitological data, both published and unpublished, were selected and integrated and how consent and sensitive information were handled is added in part 2.4.1 Data collection.
R1: The discussion section tends to repeat the findings rather than asking what they mean in a bigger sense. I would encourage the authors to reflect more critically (it does not need to be at length) on how health risks are shaped by exclusion, neglect, and uneven governance, especially in a country with the resources to do better.
Authors’ response: Thank you for your suggestion. We agree that our initial Discussion section leaned heavily on summarising results and did not sufficiently explore their broader implications. We have added a short paragraph that reflects more critically on how health risks are shaped by systemic exclusion, infrastructural neglect, and uneven governance.
R1: Minor points: acronyms like ARC and JMP should be introduced earlier, and it would be good to acknowledge potential sampling bias.
Authors’ response: The acronym JMP is introduced in the first sentence of the introduction, ARC in the third paragraph of the introduction – these are already the earliest possible moments in our narrative. We added with the first mention of the Atlas of Roma Communities: in the following also referred to as “the Atlas” for clarification. We acknowledge the possibility of sampling bias. In some settlements there was reluctance of residents to participate, and we need to rely on one time sampling. That may influence the generalizability of the findings, particularly in relation to environmental or infrastructural indicators, because we do not have ideal and representative sampling due to that problem. Added part in 4.5 Limitations.
R1: This is an important paper and has valuable contribution.
Authors’ response: Thank you very much for your appreciation, we agree and would be very glad if a larger – also non-Slovak – readership could gain access not only to our empirical study results, but particularly also to the details provided through the Atlas of Roma Communities. This very valuable resource remains underused both nationally and internationally.
R1: The English in the manuscript is generally good and understandable. It can benefit from some editing but I think the authors need to focus their attention on terminology, unclear phrasing, etc.
Authors’ response: Thank you for your comment. The language in manuscript was checked and revised.
Reviewer 2 Report
Comments and Suggestions for Authors
Comments
It is important to emphasize that living conditions can vary significantly depending on geographic location, socio-economic and cultural context, as well as the public policies implemented. Access to basic services, such as drinking water, electricity, basic sanitation, transportation, and communication, is essential to ensure well-being and quality of life. Lack of access to these services can have significant impacts on health, education, work, and natural barriers, as the authors mention, and on people's safety. Roma communities in Slovakia are not excluded from this scope.
Suggestions
The presence of an abnormal volume of intestinal parasites (which include various types of microorganisms) is the main form of contamination through the fecal-oral route from contaminated water or food. The presence of public intervention is justified. Here is a suggestion to highlight in the article.
A second suggestion is about what public authorities have been doing. Why does this situation persist? What is wrong? Why are Trichuris trichiura and Ancylostoma so common in the Community of Rome? These questions should be prominently mentioned in the article. Otherwise, we will not know what these people's responsibility is.
There are more than 100 different types of intestinal parasites that can enter the body through the nose, skin, food, water, or insect bites. Where are the health entities? This is a subject that should be explained in the article. It is the third suggestion.
Comments on the Quality of English LanguageI am not from a country where the official language is English.
Author Response
Reviewer 2
We greatly appreciate your extensive, thoughtful and detailed feedback, dear Reviewer #2. Thank you for encouraging us to rethink the and highlight more the systemic character of the situation our target group is facing, and to contextualize with (required) policies. We answered all your comments and responded to every point as listed below.
Sincerely,
The Authors
R2: It is important to emphasize that living conditions can vary significantly depending on geographic location, socio-economic and cultural context, as well as the public policies implemented. Access to basic services, such as drinking water, electricity, basic sanitation, transportation, and communication, is essential to ensure well-being and quality of life. Lack of access to these services can have significant impacts on health, education, work, and natural barriers, as the authors mention, and on people's safety. Roma communities in Slovakia are not excluded from this scope.
Authors’ response: Thank you for your comment. We agree that living conditions and access to basic services are crucial determinants and depend on geographical location, thus in the analysis we looked at different localisations of settlements. We also put some information about that in 2.1. Conceptual framing is part of the text.
R2: The presence of an abnormal volume of intestinal parasites (which include various types of microorganisms) is the main form of contamination through the fecal-oral route from contaminated water or food. The presence of public intervention is justified. Here is a suggestion to highlight in the article.
Authors’ response: Thank you for your great suggestion. We also agree that the faecal-oral route (via contaminated water and food) is a key transmission pathway for intestinal parasitic infections. Particularly when we talk about the context of inadequate WASH infrastructure. The suggestion was added in discussion. This addition further supports the justification for targeted public actions to improve WASH conditions and reduce the risk of parasitic diseases in marginalised Roma settlements.
R2: A second suggestion is about what public authorities have been doing. Why does this situation persist? What is wrong? Why are Trichuris trichiura and Ancylostoma so common in the Community of Rome? These questions should be prominently mentioned in the article. Otherwise, we will not know what these people's responsibility is.
Authors’ response: Thank you for the valuable suggestion. We agree that it is essential to explain why these parasitic infections persist and to explore the role of public authorities in addressing (or failing to address) the situation. However, in this case it is worth noting that even the small efforts of the authorities are not met with a good attitude from the Roma. The main reason is the nature of the Roma communities as a whole, who do not want to accept change, or help that would require changing their lifestyle. On the other hand, if the efforts of the authorities were better adapted to the needs of the Roma, there is a chance for change. Persistent parasitic contamination is caused by a constant cycle of infection, where even an individual cured from intestinal diseases is placed in a contaminated environment, becomes infected again and the cycle continues. The environment is contaminated in several ways, such as open defecation of residents (due to lack of sewage or water supply) or from animals (lack of control and veterinary care). Both of these points could be improved by authorities, but they are not budged friendly. The suggestion was added in discussion.
R2: There are more than 100 different types of intestinal parasites that can enter the body through the nose, skin, food, water, or insect bites. Where are the health entities? This is a subject that should be explained in the article. It is the third suggestion.
Authors’ response: Thank you for highlighting this important point. We agree that the role of public health institutions in preventing and addressing intestinal parasitic infections warrants further attention in the manuscript. In Slovakia, the Epidemiological Information System collects and analyses data on infectious diseases based on reports from healthcare professionals. However, this system primarily captures cases with clinical symptoms, as asymptomatic individuals or those who do not seek medical care are not included. As a result, despite the efforts of public health authorities, undiagnosed cases may contribute to ongoing transmission, particularly in communities with limited healthcare engagement. For example, some members of Roma communities may not participate in regular medical check-ups, further challenging effective surveillance and control. The suggestion was added in discussion.
Round 2
Reviewer 2 Report
Comments and Suggestions for Authors
Considering the amendments and corrections to the original version of the article, the document became clearer, framed, and perceptive. In any case, for future investigations and monitoring of the situation in Rome, the authors should consider the comparison with the research from this article.